# Learning-Based Hyperspectral Imagery Compression through Generative Neural Networks

**Chubo Deng, Yi Cen * and Lifu Zhang**

State Key Laboratory of Remote Sensing Science, Aerospace Information Research Institute, Chinese Academy of Sciences, Beijing 100101, China; dengcb@aircas.ac.cn (C.D.); zhanglf@radi.ac.cn (L.Z.)

**\*** Correspondence: cenyi@radi.ac.cn

**Abstract:** Hyperspectral images (HSIs), which obtain abundant spectral information for narrow spectral bands (no wider than 10 nm), have greatly improved our ability to qualitatively and quantitatively sense the Earth. Since HSIs are collected by high-resolution instruments over a very large number of wavelengths, the data generated by such sensors is enormous, and the amount of data continues to grow, HSI compression technique will play more crucial role in this trend. The classical method for HSI compression is through compression and reconstruction methods such as three-dimensional wavelet-based techniques or the principle component analysis (PCA) transform. In this paper, we provide an alternative approach for HSI compression via a generative neural network (GNN), which learns the probability distribution of the real data from a random latent code. This is achieved by defining a family of densities and finding the one minimizing the distance between this family and the real data distribution. Then, the well-trained neural network is a representation of the HSI, and the compression ratio is determined by the complexity of the GNN. Moreover, the latent code can be encrypted by embedding a digit with a random distribution, which makes the code confidential. Experimental examples are presented to demonstrate the potential of the GNN to solve image compression problems in the field of HSI. Compared with other algorithms, it has better performance at high compression ratio, and there is still much room left for improvements along with the fast development of deep-learning techniques.

**Keywords:** compression sensing; generative neural network; hyperspectral; feature reduction

## 1. Introduction

Hyperspectral remote-sensing technology arose in the 1980s; it can obtain many very narrow and continuous image data in visible, near-infrared, medium, and long-wave infrared spectra. The data collected by these instruments help us revolutionize our understanding of climatology, meteorology, and land management. Hyperspectral images (HSIs) typically possess a high degree of spectral, as well as spatial, correlation. Moreover, remote-sensing applications require the collection of high volumes of image data, of which HSIs are a particular type [1]. HSI compression can significantly reduce onboard memory requirements, communication channel capacity, and download time. Depending on the quality of the reconstruction required, compression algorithms can be either lossless or lossy [2–3]. Dimensionality reduction methods [4–6] provide approaches to deal with the computational difficulties of HSIs; these approaches usually use projections to compress a high-dimensional data space into a lower-dimensional space using multiplication by a coefficient matrix. For lossy compression [7,8], the image reconstructed from the lower-dimensional space is not exactly the same as the original data.

Wavelet-based [9,10] lossy compression techniques are of particular interest because of their long history of providing excellent compression performance for traditional two-dimensional (2D)

imagery. Consequently [11], several prominent 2D compression methods have been extended to 3D; these 3D wavelet-based techniques include 3D SPIHT (Set Partitioning in Hierarchical Trees), 3D SPECK (set partitioned embedded block), and 3D tarp [12]. Moreover, the JPEG2000 [13–16] standard has been widely applied to 3D HSI coding (e.g., [13]) because of its ability to code multiple image components. However, it has been argued that such a direct extension from 2D to 3D without the consideration of the special characteristics of HSIs may be problematic [14], because data analysis applied subsequent to compression may be affected. Typically, a 3D compression algorithm involves coupling a decorrelating transform in the spectral dimension with a spatial wavelet transform plus a coding algorithm suitably modified for a 3D data array. Most often, a wavelet transform is also deployed spectrally to implement a 3D wavelet decomposition [17]. Indeed, the JPEG2000 standard supports a spectral discrete wavelet transform in this manner, and it has been shown that JPEG2000 plus a spectral discrete wavelet transform generally achieves a rate distortion performance superior to that of other wavelet-based techniques.

Principle component analysis (PCA) is another popular decompression technique [18]; PCA extracts important information from the data to represent it as a set of new orthogonal variables called principal components. As a result, the original data can be reconstructed by these principal components by multiplication with an inverse transform matrix. However, PCA may not be the optimal choice for the feature reduction task, since both the PCA transform and the inverse transform are linear, so it cannot extract information with nonlinear relationships.

In general, most feature reduction methods other than deep learning are linear transforms, such as those that use geometric structure [19] to reduce the number of spectral bands and preserve valuable intrinsic information in the HSI or approaches based on the wavelet transform to reduce the dimensional space. In mathematics, most familiar transforms that are used as tools for feature reduction are linear, such as the Fourier transform [20], PCA transform, wavelet transform [21], and Laplace transform [22], because nonlinear space is much more difficult to handle theoretically than linear space. Hence, until the emergence of neural networks, most proposed algorithms used a linear transform or a linear transform with some nonlinear modifications.

Neural networks are a set of algorithms that mimic the human brain and are designed to recognize patterns. In contrast, the human brain can not only accomplish the recognition task but is also able to memorize words, images, and sound. Many studies have been performed on the recognition task using neural networks, but very few of these studies focus on their ability for storage. Neural networks take advantage of probability theory and the back-propagation method, making nonlinear problems solvable. In this work, a generative neural network (GNN) approximates a continuous function as much as possible, which yields a mapping between the latent space and the data distribution space.

In this paper, we find that the image can be generated directly from the random latent code. The latent code is a representation of compressed data [23,24]. The GNN builds a mapping between the latent space and the image space. Theoretically, the generated image can be as close as possible to the original one if the random latent code, and the GNN can provide enough degrees of freedom. The regular compression methods do not work well with a high compression ratio, but the GNN improve such ability. In our experiment, we compare the result from our method with the 3D SPECK and 3D SPIHT algorithms, and our method is more accurate than the other two algorithms, especially at large compression ratios. Since we provide a novel way to compress a HSI, the compression ratio needs to be redefined as the size of the original HSI to the size of the neural network.

The abbreviation letters GNN are famous for graph neural network in the current deep-learning field; many researches have explored novel algorithms through graph convolution on HSI, such as [25]. However, in this paper, GNN stands for generated neural network, which is different from graph neural network.

## 2. Methodology

The real data distribution of HSI $p_r$ admits a density, and it is supported by a low-dimensional manifold. Rather than estimating distribution $p_r$, we define a random variable $X$ with a fixed

distribution $p(x)$ and pass it through a parametric function $g_\theta : X \rightarrow Y$ (the GNN) that directly generates the HSI following some distribution $p_\theta$. By changing parameter $\theta$, we can change the distribution and make it as close as possible to the real distribution $p_r$.

That is, $\forall \varepsilon > 0$; there exists a positive integer $M$, such that when the number of iterations of the GNN $n > M$, we have

$$|H(P_r) - H(P_\theta)| < \varepsilon \tag{1}$$

The difference between the distributions is measured by the entropy, denoted as H, which is given by the following formula [26]

$$H(x) = -\int p(x)\log(p(x))dx \tag{2}$$

For a particular image, there is a finite number of pixels and bands; hence, in a real experiment, the discrete form of entropy is used, which is expressed as follows:

$$H(x) = -\sum_{i=0}^{N} p_i(x)\log(p_i(x)) \tag{3}$$

In our experiment, a HSI is normalized to the range 0–255 for the convenience of display, and in this case, integer N is equal to 255. Probability density $p_i(x)$ is calculated as the number of pixels with a specific integer value divided by the total number of pixels in the HSI. Note that the discrete formula in Equation (3) does not converge to Equation (2) when N $\rightarrow \infty$.

A neural network is a computer program that learns from the data: it can extract feature information from images, and a well-trained network can contain all the information that is used for regression, classification, or even pixel-level object detection tasks. For example, the famous VGG16 [27] network uses convolution and pooling layers, mapping an image of size 224 × 224 × 3 to 7 × 7 × 512. This is then followed by two fully connected layers to obtain the classification result, which is in the form of a one-hot encoding vector. A well-trained VGG16 network is a representation of the whole dataset. In this paper, we reconstruct the images from a neural network that can be considered as a reverse VGG16 structure that maps a single dimensional latent code to the HSI via convolution and upsampling layers.

## 2.1. Architecture of the GNN

Typically, compression algorithms utilize statistical structure in the images. In a HSI, there are two types of correlations. One is spatial correlation, which exists between neighborhood pixels in the same band; the other is spectral correlation, which exists in the adjacent bands. The architecture of a GNN is designed for compression using both spectral and spatial correlations.

Initially, we considered the variational autoencoder(VAE) [28] as a suitable structure for our compression task; The structure of VAE is displayed in Figure 1, it inherits the traditional approach to compression by encoding and decoding processes through a neural network. The HSI can be encoded as a one-dimensional (1D) latent code, and the latent code can be decoded without any loss if the autoencoder system provides enough degrees of freedom. The autoencoder can be solely trained to encode and decode with as few degrees of freedom as possible, no matter how the latent space is distributed. It is natural for the network to take advantage of the potential for overfitting to achieve its task because it is able to do so. Usually, overfitting is a modeling error that occurs when a function is too closely fit to a limited set of data points. In our task, overfitting will lead to a perfect match between the original image and the reconstructed one, which is what we desire.

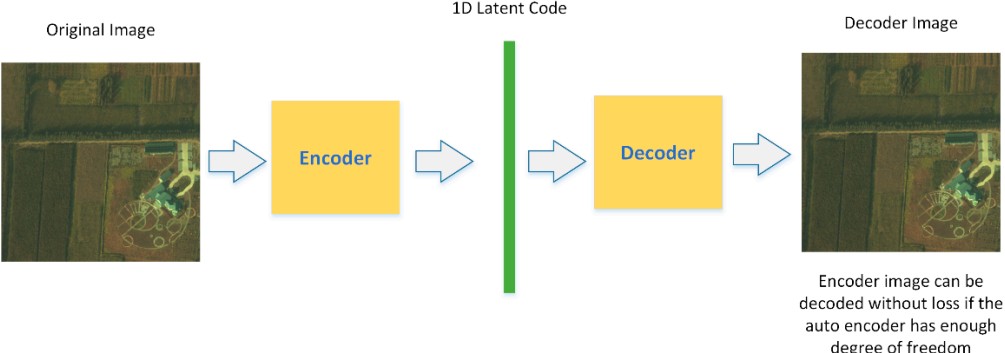

**Figure 1.** Variational autoencoder architecture.

In Figure 2, the architecture of the GNN is shown. The process starts from a random latent code, which is generated by random normal distribution with vector length 50. Then, this short 1D vector is amplified to a long vector by a fully connected layer to a size that matches the first cube in Figure 2. The small cube is passed through a stack of convolutional layers, where we use filters with a very small receptive field: 3 × 3. Then, followed by an activation function ReLU (Rectified Linear Unit) and the bilinear upsampling layer, the small cube is amplified to double the size in spatial dimension and has a channel equal to the number of filters in the convolutional layer; each block consists of one convolutional layer, one activation function, and one upsampling layer. Through multiple blocks, the initial cube is finally extended to the original HSI cube. Certainly, the architecture is not a unique one that solves the problem; it is flexible enough to implement the convolution and upsampling blocks with different sizes and simply reuse the same structure multiple times when defining the forward pass, and the compression ratio is directly determined by the number of convolution and upsampling blocks. However, this is a more flexible choice than traditional compression methods, because we can manipulate the structure of the network to determine the quality of the generative HSI.

Since we focus on the compression task, it would be meaningless if the number of neurons in the GNN were larger than the number of nodes in the HSI. Hence, the upsampling and convolution blocks are necessary in the network. The upsampling block is used for compression and interpolation using the spatial correlation, and the convolution block is used for interpolation using the spectral correlation. The channels in the neural network are treated as the representation of spectral information, which yields the features of the HSI.

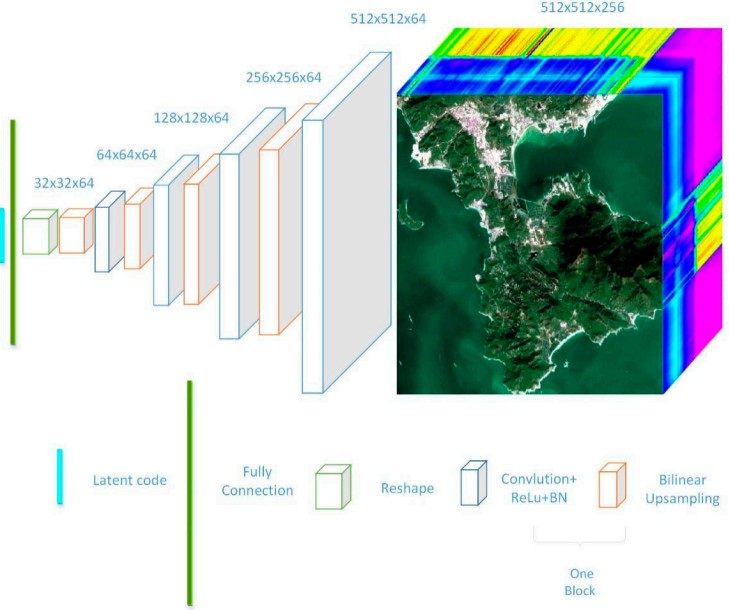

**Figure 2.** Generative neural network (GNN) architecture.

### 2.1.1. Bilinear Interpolation for Upsampling

For spatial correlation, we utilize a bilinear interpolation for upsampling [29] from a lower-dimensional space to a higher one. The scheme used for bilinear interpolation is illustrated in Figure 3, where the points $Q$ are four adjacent pixels, and P is the point we want to evaluate.

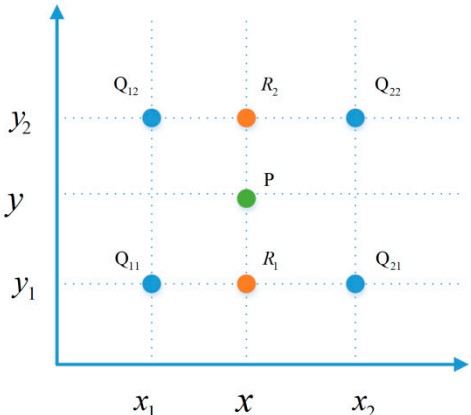

**Figure 3.** Bilinear interpolation.

First, the values at two points, $R_1$ and $R_2$, are computed by linear interpolation, which is

$$f(R_1) \approx \frac{x_2 - x}{x_2 - x_1} f(Q_{11}) + \frac{x - x_1}{x_2 - x_1} f(Q_{21}) \tag{4}$$

$$f(R_2) \approx \frac{x_2 - x}{x_2 - x_1} f(Q_{12}) + \frac{x - x_1}{x_2 - x_1} f(Q_{22}) \tag{5}$$

Next, point $P$ is given by a linear interpolation of $R_1$ and $R_2$ as follows:

$$f(P) \approx \frac{y_2 - y}{y_2 - y_1} f(R_1) + \frac{y - y_1}{y_2 - y_1} f(R_2) \tag{6}$$

Bilinear interpolation outputs an approximation for a point that is better than taking the average of the neighbor points or simply copying the value of an adjacent point. It, hence, more accurately estimates the upsampling pixels in the HSI in each channel in the network.

### 2.1.2. Latent Code and Entropy

The whole generative process starts with a latent code, and the latent code is a 1D normally distributed vector. In Figure 2, we illustrated how the 1D latent code is first extended to a long vector through a fully connected layer and then reshaped into a 3D cube. Since our final target is a 3D HSI, it is natural to use a 1D vector to start the generation process instead of a 2D matrix. In the network architecture, a 2D matrix can be directly used in the upsampling and convolution layers, and it seems that a 2D matrix would be more suitable for the GNN as the starting code. However, it is not necessary to use a 2D matrix as the initializer, because a 2D random normally distributed matrix provides the same information as a 1D random normally distributed vector measured by the entropy defined in Equation (3). We use an example to demonstrate this statement.

Consider a random normally distributed matrix, $A^{n \times m}$ with m < n; we calculate the entropy of the matrix $A^{n \times m}$, and the entropy H(A) remains almost constant when m is varied from 1 to 1000 for the fixed n, as shown in Figure 4. The small fluctuation is due to the discretized version of the entropy calculation and numerical error. When we set m to be one, the matrix A becomes a 1D normally distributed vector, which means a 1D normally distributed vector can provide the same entropy information as a 2D matrix.

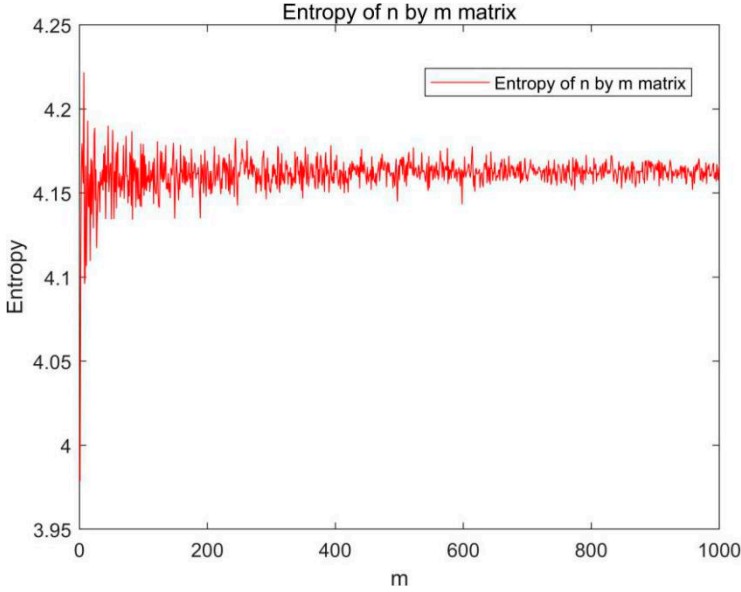

**Figure 4.** Comparison of entropy for a n × m matrix for a fixed n = 1000.

A theoretical explanation is as follows: the columns in the normally distributed matrix $A^{n \times m}$ are not linearly independent, and the summation of each column in matrix $A^{n \times m}$ is equal to zero; therefore, each additional column can be represented by the existing columns, so the rank of this N × M matrix is only one. This is the reason that it is preferable for most GNNs to use a 1D latent code; it plays the same role as a 2D code but leads to less computational cost. The next question that arises is why we use a normal distribution instead of another distribution. This is because the normal distribution theoretically provides the maximum entropy [30,31]. The proof is as follows:

The entropy optimization problem can be stated as

$$\text{Max} \int_{-\pi}^{+\infty} p(x) \log(p(x)) dx \tag{7}$$

with constraints

$$\int_{-\infty}^{+\infty} p(x) dx = 1$$
$$\int_{-\infty}^{+\infty} x p(x) dx = \mu \tag{8}$$
$$\int_{-\infty}^{+\infty} (x - \mu)^2 p(x) dx = \sigma^2$$

The maximum entropy problem is given by Equation (7), with constraints on the probability density function, mean, and variance in Equation (8). We employ the Lagrange multipliers $\alpha$, $\beta$, and $\gamma$ on Equation (7), where the Lagrange equation is given by

$$\begin{aligned} L(p, \alpha, \beta, \gamma) &= \int_{-\infty}^{+\infty} p(x) \log(p(x)) dx \\ &+ \alpha (\int_{-\infty}^{+\infty} p(x) dx - 1) \\ &+ \beta (\int_{-\infty}^{+\infty} x p(x) dx - \mu) \\ &+ \gamma (\int_{-\infty}^{+\infty} (x - \mu)^2 p(x) dx - \sigma^2) \end{aligned} \tag{9}$$

By taking the derivative of Equation (9) with respect to p and setting it to zero, we have the following:

$$\frac{\partial L}{\partial p} = -\frac{\partial}{\partial p}(\int_{-\infty}^{+\infty} p(x)\log p(x) - \alpha p(x) - \beta x p(x) - \gamma(x-\mu)^2 p(x)dx) = 0 \qquad (10)$$

Let

$$F = p(x)\log p(x) - \alpha p(x) - \beta x p(x) - \gamma(x-\mu)^2 p(x) \qquad (11)$$

By functional analysis, we can have $\frac{\partial F}{\partial p} = 0$. That is,

$$\frac{\partial F}{\partial p} = \log p(x) + 1 - \alpha - \beta x - \gamma(x-\mu)^2 = 0 \qquad (12)$$

One can solve it as

$$\begin{aligned} p(x) &= e^{\lambda-1}e^{\alpha x + \beta(x-\mu)^2} \\ &= Ce^{\beta(x-\mu+\frac{\alpha}{2\beta})^2} \end{aligned} \qquad (13)$$

Applying the constraints in Equation (8), we can solve it as follows:

$$p(x) = \frac{1}{\sqrt{2\pi}\sigma} e^{-\frac{(x-\mu)^2}{2\sigma^2}} \qquad (14)$$

which is the normal distribution density function. This proves that a normally distributed latent code provides a higher entropy than any other distribution.

### 2.1.3. Model Weight Pruning

Weight pruning [32] is another core technique used in our GNN that substantially increases the compression ratio, with only a limited loss of accuracy. Its aim is to eliminate unnecessary values in the weight tensor, reduce the number of connections between neural network layers and calculations, and, thus, reduce the number of operations. Figure 5 illustrates how weight pruning works, where the unimportant neurons and their connections to other neurons are deleted.

For a particular task, we need to find the optimal way to reduce the weight nodes without substantially impacting the accuracy. The compression of the HSI is equivalent to the compression of the neural network, since the neural network is the representation of the HSI. Moreover, we can take advantage of the structure in the network, because the weights in the fully connected layers usually make up a large percentage of the weights in many networks. Take the VGG16 network, for example, in which over 90% of the parameters are concentrated in the fully connected layer. In our case, the situation is the same. The first layer is a fully connected layer, which has a large percentage of weights. Assume the number of deleted neurons is K. Then, K × J neurons are removed from the fully connected layer, where J is the number of neurons in the next layer. Hence, we first attempt the pruning at the first layer. The neuron ranking used for deletion in this layer is fairly simple. We use the L1 norm of the weights of each filter. At each pruning iteration, we rank all the weights, prune the K-lowest ranking weights in this layer, retrain the network, and repeat. The number of lowest ranking weights can be chosen manually to control the compression ratio and the quality of the generated image.

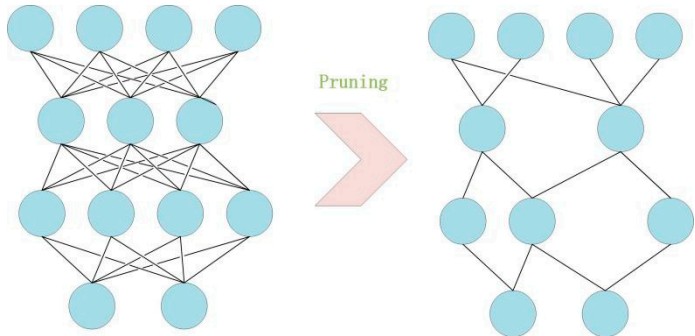

**Figure 5.** Weight pruning demonstration.

## *2.2. Multiple Images and Huge Image Training*

### 2.2.1. Embedding and Multi-Image Training

Since a neural network starts from a random latent code, the generated random codes are different for each experiment. Surprisingly, different random codes can generate a unique image under the same structure of the generated network, because they follow the same distribution. This is the reason that a trained network can be used to store an image even if the inputs are different each time. However, when we consider the storage of multiple images, this convenience leads to some difficulties, because there is no designated label for each image, so only one image can be generated for each training. This problem can be resolved by the embedding technique.

The embedding technique [33] is frequently used in natural language processing. It transforms a word into a unique vector that can be used for computation. Here, we embed an integer into a random normal distribution so that the normal distribution is no longer random but unique. Since the generated HSI is uniquely determined by this given integer, the image can be encrypted by it. Without knowing the specific integer, the receiver can no longer generate this HSI, and this property can be used to transport classified HSIs.

Figure 6 illustrates embedding for an integer sequence from 0 to 6 with random vectors, such that one group of points can be distinguished from the other groups [34]. Although each point in a group has a different value from the others, they are clustered in a neighborhood area. The difference in each group will eventually become the difference in each HSI generated through the GNN. The embedding technique makes it possible to train for multiple images. Since the input becomes a $X \times Y$ random normally distributed matrix with $X$ samples and vectors of length $Y$, our neural network should be able to use it to output $X$-designated images.

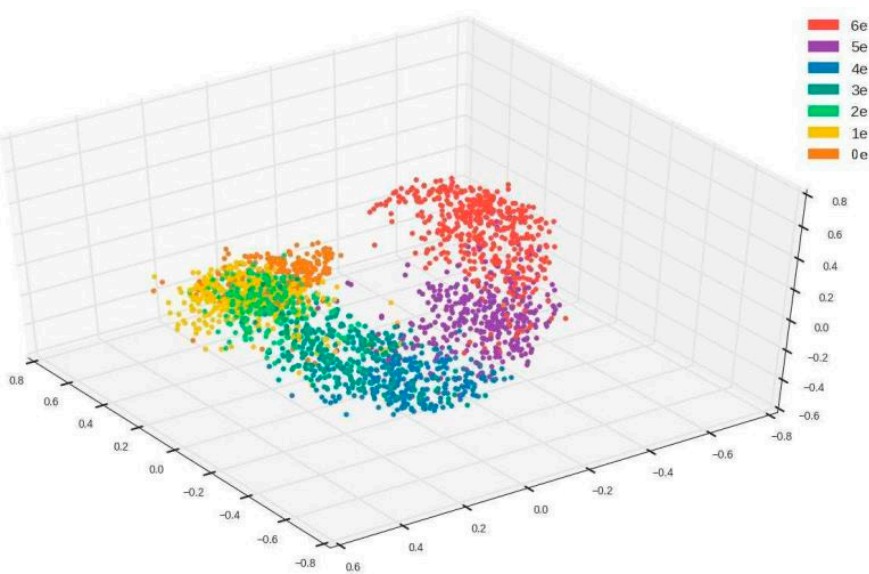

**Figure 6.** Illustration of the embedding technique.

### 2.2.2. Huge Images and Batch Training

As high-resolution instruments over a very large number of wavelengths have been developed, the HSIs collected by such instruments have begun to occupy huge amounts of storage. This leads to the practical problem that the memory in a GPU (Graphic Processing Unit) is not sufficient for our training. When it comes to a custom deep-learning task in which a huge dataset would overload a GPU, batch training [19] can be used, which trains one batch each time to reduce the burden on the GPU memory. However, in our task, we only trained one huge image. Under such circumstances, we can split the image into several small blocks and train some of them in a batch, which can substantially reduce the consumption of GPU memory.

Figure 7 gives an example of how the image is divide into different blocks and trained in batches. The number of blocks and the batch size are determined by the capacity of the GPU storage. The generated HSIs are then used to reconstruct the original image according to the order of the given embedding integers. Batch training cannot achieve the same quality as single-image training with the same number of epochs, but we can train for more epochs to minimize the loss function to acquire the same-quality images. Training using more epochs compensates for the shortage of GPU capacity.

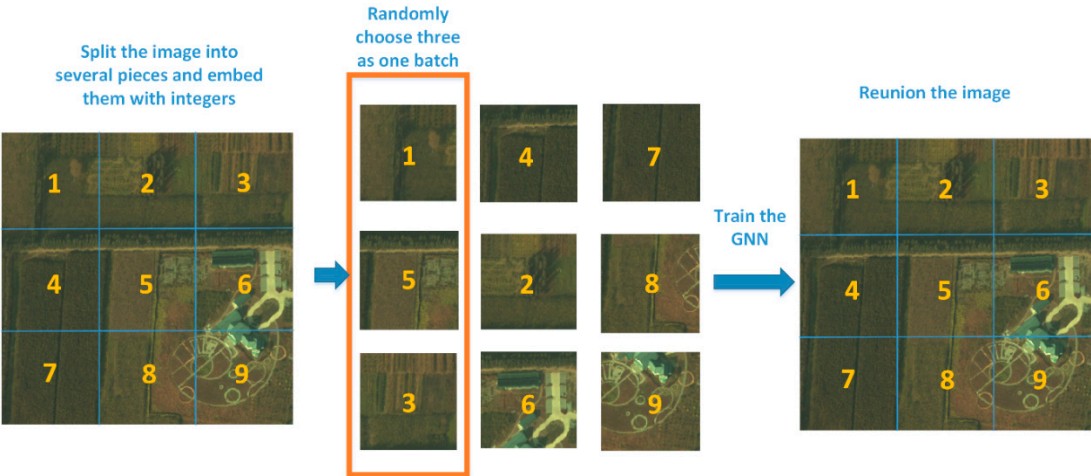

**Figure 7.** Batch training for a huge image.

## 3. Experimental Result

### 3.1. Data and Implementation Description

The main dataset was collected by the authors, and the images were acquired by a newly designed airborne hyperspectral sensor accompanied by synchronous ground survey experiments. The flight height was 2000 m, and the flight areas covered Xiong County, An County, Rong County, and Baiyangdian Lake in Hebei, China. The east–west length of this region is 48 km, its north–south width is 27.5 km, and its total area is 1320 km². The spectral range of the HSI of Xiongan New Area (Matiwan Village) is from 400 to 1000 nm, with 256 bands and a spatial resolution of 0.5 m. The image size is 3750 × 1580 pixels. The whole image is displayed in Figure 8.

The image in Figure 8 covers a very large area, and it is difficult to see small details without magnification. In the next section, we present the results of our experiment by selecting a part of the image to enable a clear and detailed comparison of the generated and original images. Our training was implemented on a single GPU NVIDIA GTX 1060 5G. It was carried out by optimizing the power loss function using the gradient descent with Adam [355]. Since only one image was fed to the network in our training, the batch size was set to one. We stopped the training after 15k epochs. The learning rate was set to $5 \times 10^{-4}$. There was no dropout layer, because we did not treat overfitting as a problem.

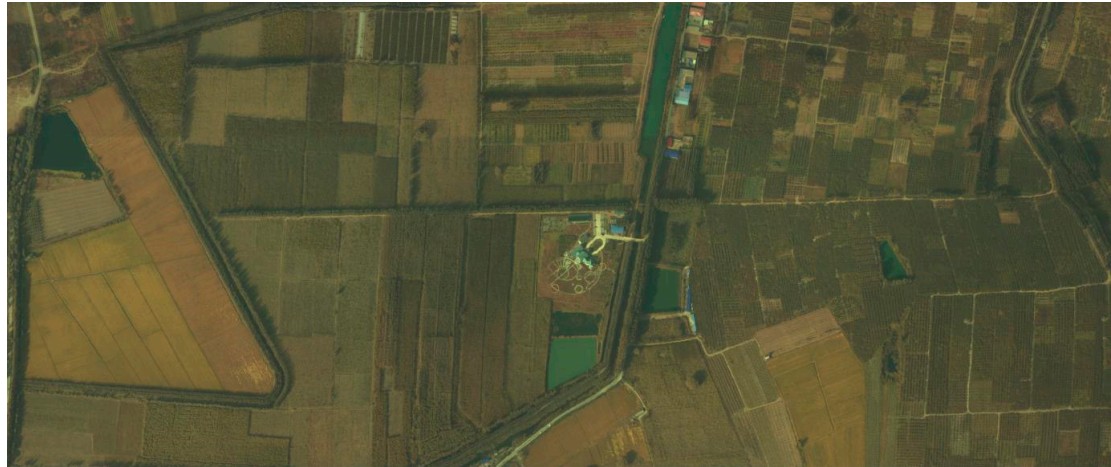

**Figure 8.** True color image of Xiong An New Area, Hebei, China (Matiwan Village).

### 3.2. Generated Image Sequence Demonstration

In the following sections, we demonstrate the evolution of the generated images at different epochs. They illustrate how the images are generated from a random chaos image to a highly organized HSI. In Figure 9, the generated image starts with random noise. After one hundred epochs, one can already see the outline of the original structure but few details, and there are many large speckles in the image. As the iterations continue, the speckles become smaller, and the contrast increases. When the iterations finish, we can barely see a difference between the generated and original images. The network successfully generated the original HSI, and, hence, it can be used to store the HSI.

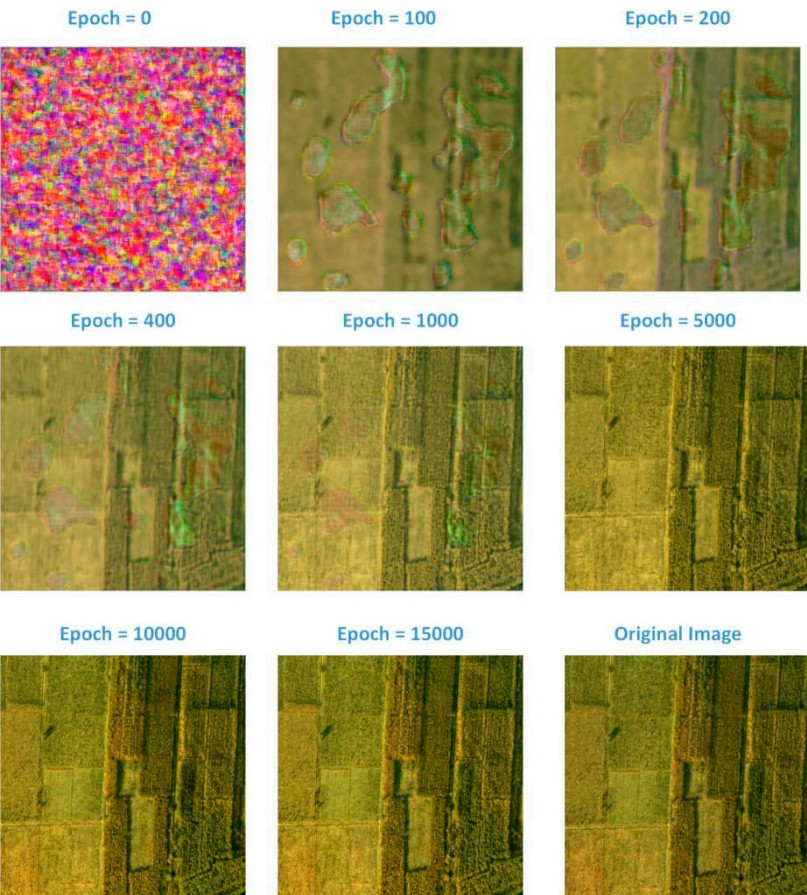

**Figure 9.** True color-generated hyperspectral images (HSI) 512 × 512 pixels in size at different epochs.

Figure 10 gives an intuitive display for a single-pixel approximation in the spectral dimension, which shows how close the generated pixel approximates the original one. Training was performed using a 512 × 512 × 256 HSI that was normalized to a range from −1 to 1.

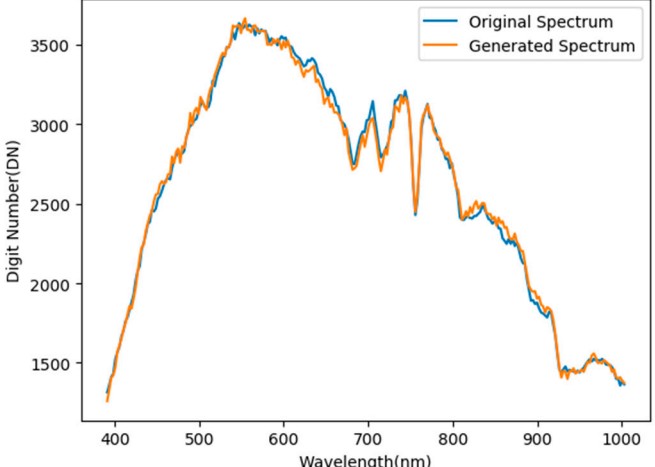

**Figure 10.** Comparison of generated and original pixels in the spectral domain.

The final training loss was $2.7 \times 10^{-4}$ and was measured by mean square error (MSE) using our modified loss function. Initially, the MSE was used during training as a loss function. However, we observed that it was not the optimal metric after several experiments. We employed a loss function that was a power function of the difference between the original and generated HSIs, which was given by

$$ loss = \sum \left| y_{pred} - y_{true} \right|^{p} \tag{15}$$

This is called the $l^{p}$ norm in mathematics, and it measures the differences between two functional spaces. Table 1 presents the $l^{p}$ norm loss under different values of $p$. Since they are obtained under different measurements, the $l^{p}$ norm loss cannot be used for comparison; therefore, we regularize the error to the mean square error for comparison purposes.

**Table 1.** Comparison of the mean square error (MSE) for different values of $p$ at the 15-thousandth epoch.

| Criteria | p = 1 | p = 1.5 | p = 2 | p = 3 |
|---|---|---|---|---|
| MSE | $3.25 \times 10^{-4}$ | $2.69 \times 10^{-4}$ | $5.83 \times 10^{-4}$ | $5.75 \times 10^{-4}$ |
| $l^{p}$ (norm loss) | $1.212 \times 10^{-2}$ | $1.87 \times 10^{-3}$ | $5.83 \times 10^{-4}$ | $3.21 \times 10^{-5}$ |

Figure 11 presents the change in loss after each iteration for different values of power $p$. We observed a fast and steady convergence when $p$ was 1.5. The loss settled down after 6000 epochs of training to a number below $5 \times 10^{-4}$ and, finally, converged at around $2.7 \times 10^{-4}$. In contrast, other values of $p$ did not result in such a fast convergence rate, and sometimes, the loss became extremely large and unstable. For a small value of $p$, we observed a singularity point in the generated image, even if the total loss was small.

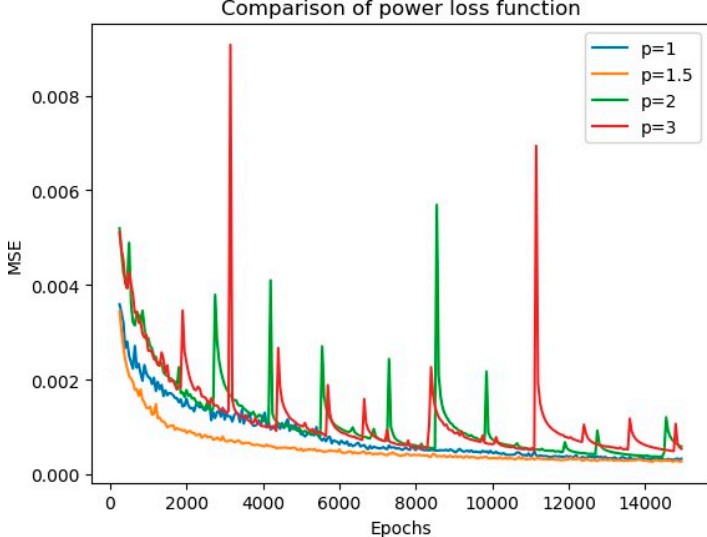

**Figure 11.** Comparison of the power loss function measured using the mean square error (MSE) criterion.

Figure 12 shows a black point in the left bottom corner of the generated HSI that severely impacts the visual quality, even though the total loss is small. This phenomenon occurs, because a large difference in some pixels is significantly reduced by a power function with a low value of *p* compared with those with a high value of *p*. The total loss is an average of difference for all pixels, so a big difference at this singularity point will be reduced by the loss function and averaged to a very small number [36]. Hence, the backwards propagation process does not perform enough corrections for this singularity point. Therefore, it is necessary to find a value of *p* that balances the correction of this error, while ensuring that the loss is as small as possible.

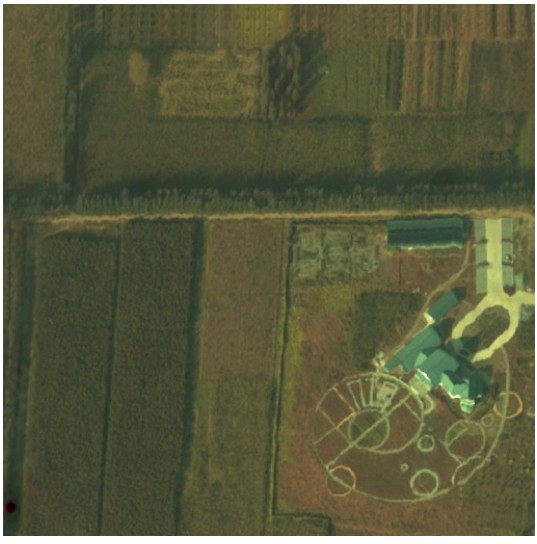

**Figure 12.** Singularity point occurring with a small value of *p*.

*3.3. Huge-Image Training Experiments*

As described in the previous section, a huge HSI occupies a large storage space, and the corresponding GPU will exhaust its memory resources when training the neural network. To fix this problem, we implemented the idea of batch size training. The huge HSI is first split into several blocks, and then, only some blocks are fed to the GPU at each epoch. After the training, we can generate the HSI by inputting the embedded latent code to the neural network and then reconstruct the HSI according to this embedded code. However, the experiments showed that the MSE was larger than that of the single-image training, even with the additional iterations.

In Figure 13, the HSI is first split into four blocks, and we randomly selected two of them for training at each batch. In the beginning, the reconstructed image consists of four random noise images, and the boundaries are obvious. Then, four almost indistinguishable images are generated by the network, and this phenomenon draws our attention. Although each block of the image is vague, it contains information about the whole HSI. For example, some details appear in the bottom-right block that should belong to the top-left block at epoch 400. As the iterations continue, we observe that the boundary of each block becomes vague and more details appear. Finally, the boundaries disappear, and the reconstructed image is an approximation of the original image.

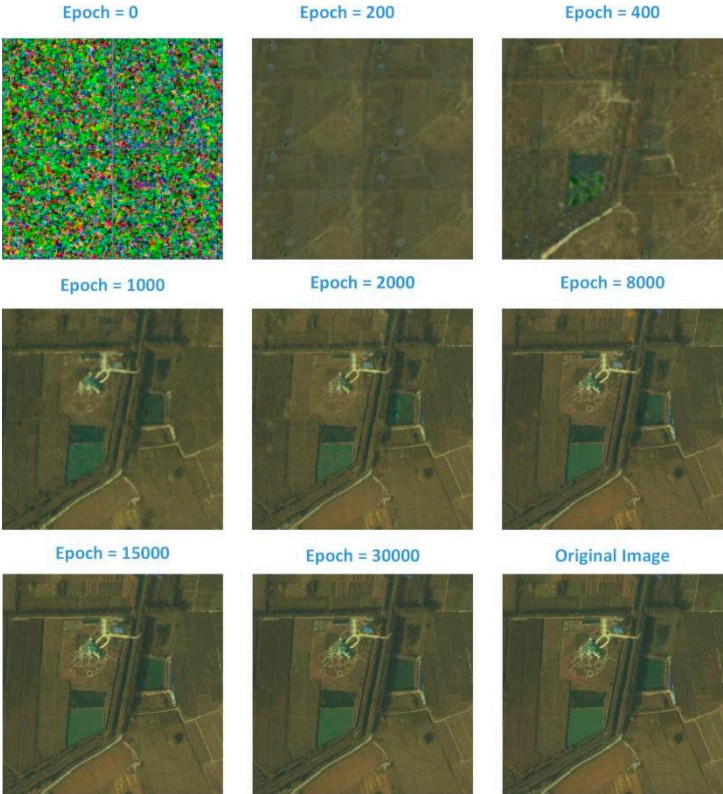

**Figure 13.** Batch training demonstration.

In Figure 14, the loss function of the batch-size training is presented for a huge image, and we observe that the MSE decreases to around 0.001 at 15k epochs and oscillates between 0.001 and 0.002, which is substantially larger than the loss for single-block image training. As a result, the batch training is only a compromise for reducing the burden on GPU storage.

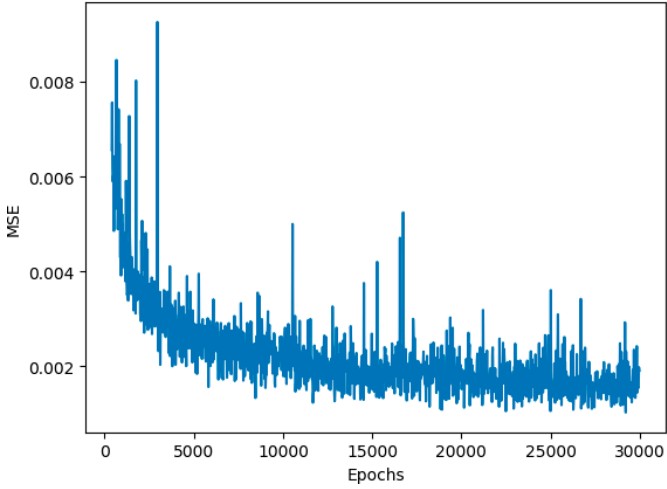

**Figure 14.** Loss of batch-size training.

*3.4. Comparison of Compression Capacity*

In this paper, we not only used MSE as an evaluation criterion but, also, employed the peak signal-to-noise ratio (PSNR) and compression ratio as criteria, which are common standards of quality in image compression. The neural network stores all the information and is used to generate the HSI, and it directly determines the compression quality. The PSNR is an engineering term for the ratio between the maximum possible power of a signal and the power of the corrupting noise that affects the fidelity of its representation. The PSNR is defined as follows:

$$PSNR = 10\log_{10}(\frac{2}{MSE}) \tag{16}$$

Different structures of GNNs give different compression rates and quality. The core structure consists of several blocks. A basic block includes one upsampling layer, one batch-normalization layer, and one ReLU activation function layer. The loss function for different structures are demonstrated in Figure 15. The upsampling layers are the most important layer in the block, because they amplify the feature map to a fixed ratio with a default value of two. We use bilinear interpolation for upsampling to better approximate the next layer. Without the upsampling layer, there is no compression.

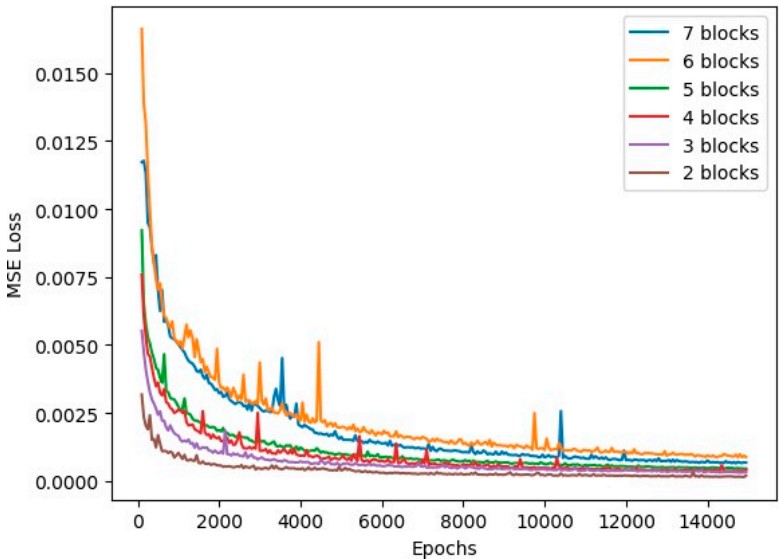

**Figure 15.** Performance comparison of different GNN structures.

Table 2 shows the performance of the proposed method for different numbers of blocks. The data shows that more blocks lead to a higher compression rate but lower the generated image quality. It is reasonable to expect this phenomenon, because the upsampling layer reduces the number of parameters, and at the same time, it approximates the adjacent pixels by a bilinear interpolation algorithm, which creates a difference between the generated image and the original one. Hence, a compromise between the compression ratio and the quality must be made, according to the requirements.

**Table 2.** Compression quality comparison for different generative neural network (GNN) structures before and after pruning. PSNR: peak signal-to-noise ratio and CR: compression ratio.

| Criteria | 2 Blocks | 3 Blocks | 4 Blocks | 5 Blocks | 6 Blocks | 7 Blocks |
|---|---|---|---|---|---|---|
| MSE | $1.46 \times 10{-4}$ | $3.05 \times 10^{-4}$ | $3.56 \times 10^{-4}$ | $4.50 \times 10^{-4}$ | $8.71 \times 10^{-4}$ | $6.44 \times 10^{-4}$ |
| PSNR | 44.38 | 41.17 | 40.50 | 39.48 | 36.62 | 37.92 |
| Size of GNN | 207MB | 54.7MB | 16.4MB | 7.04MB | 4.79MB | 4.34MB |
| CR | 0.66 | 2.50 | 8.35 | 19.46 | 28.60 | 31.57 |
| MSE (After pruning) | $1.48 \times 10^{-4}$ | $3.09 \times 10^{-4}$ | $3.67 \times 10^{-4}$ | $4.95 \times 10^{-4}$ | $1.01 \times 10^{-3}$ | $9.32 \times 10^{-4}$ |
| Size of GNN (After pruning) | 72.5MB | 20.7MB | 7.38MB | 4.9MB | 4.31MB | 4.15MB |
| CR (After pruning) | 1.89 | 6.62 | 18.56 | 27.96 | 31.78 | 33.01 |

The weight pruning technique leads to further compression in the network by reducing the number of weights. Since we only apply the pruning at the first fully connected layer, it leads to different results for different structures. With a few blocks, the first layer contains a large percentage of the total number of parameters, so that the pruning leads to good results while barely reducing the accuracy. On the contrary, the pruning becomes less efficient with more blocks, because the first layer does not have a large portion of the total number of parameters. For instance, in our experiment, we deleted two-thirds of the weights in the first layer and left one-third remaining. In a structure with only two blocks, the first fully connected layer contains 98% of the parameters, which reduces to 77% for four blocks. The percentage of deletion decreases as the number of blocks increases.

Furthermore, we also compared our results from GNN with the 3D SPECK and 3D SPIHT algorithms in Table 3. Since our network structure cannot support an arbitrary compression ratio, we chose the closest compression ratio for comparison. The left number in the first column is the compression ratio of the GNN, and right number is the compression ratio for the other two algorithms. Our proposed GNN performs better when the compression ratio is large, but it does not work well with lower compression ratios. Since we do not employ the pruning technique in the convolution block, there is still plenty of potential for further decreasing the compression ratio. Moreover, many techniques in deep learning, such as residual blocks, can be employed to increase the accuracy of the GNN.

**Table 3.** Performance comparison for different algorithms using PSNR criteria.

| Compression Ratio | GNN | 3D SPECK | 3D SPIHT |
|---|---|---|---|
| 31.78/32 | 36.62 | 28.91 | 30.16 |
| 18.56/16 | 40.50 | 36.34 | 37.49 |
| 6.62/6 | 41.17 | 48.83 | 49.55 |

*3.5. Comparison of Different HSIs*

Three other open HSIs were used to test our GNN to understand its performance on different HSIs. They are the Pavia Center, Botswana, and Washington DC Mall. The Pavia Center consists of 102 available spectral bands and is a 1096×1096 pixel image acquired by the ROSIS sensor during a flight campaign over Pavia. For Botswana, the Hyperion sensor on EO-1 acquired data at a 30-m pixel resolution over a 7.7-km strip with 242 bands covering the 400–2500-nm portion of the spectrum in 10-nm windows. Preprocessing of the data was performed by the UT Center for Space Research to mitigate the effects of bad detectors, inter-detector miscalibration, and intermittent anomalies. Uncalibrated and noisy bands that covered water absorption features were removed, and the remaining 145 bands were included as candidate features. In the Washington DC Mall image, there were 210 bands in the 0.4 to 2.4-μm region of

the visible and infrared spectrum. This dataset contained 1208 scan lines with 307 pixels in each scan line. It totaled approximately 150 MB.

The Pavia Center image was cropped to a size of 512 × 512 pixels to the equal size used in the previous experiments. The Botswana and Washington DC Mall images were cropped into a size of 512 × 256 pixels, because their widths were smaller than 512. The cropped Pavia Center, Washington DC Mall, and Botswana images had sizes of 42 MB, 32 MB, and 27 MB, respectively.

We present the MSE, compression ratio, and PSNR performances of the GNN for different HSIs with three and four blocks in Table 4. The left column of each criterion presents the results for three blocks, and the right column is for four blocks. The observations acquired by these experiments reveal that the MSE does not change much for the Matiwan Village, Pavia Center, and Botswana HSIs with respect to structure. In this case, we will naturally choose four blocks as our structure, since it yields a higher compression ratio than the three-block structure, without a loss of accuracy. Moreover, the cropped size of Matiwan Village (137 MB) is much bigger than those of the other three HSIs, since it contains more bands in the spectral dimension and more pixels in the spatial dimension than the others. Hence, our GNN can obtain a higher compression rate. This is because we can expect the compression in the spectral dimension to be a more significant contribution to the overall compression when the image includes more spectral bands.

**Table 4.** Performance comparisons on different hyperspectral images (HSIs) with three and four blocks without pruning.

| Name of HSI | MSE | | Compression Ratio | | PSNR | |
|---|---|---|---|---|---|---|
| MATIWAN VILLAGE | $3.05 \times 10^{-4}$ | $3.56 \times 10^{-4}$ | 2.50 | 8.35 | 41.17 | 40.51 |
| PAVIA CENTER | $3.47 \times 10^{-4}$ | $3.59 \times 10^{-4}$ | 1.45 | 4.22 | 40.62 | 40.47 |
| DC MALL | $1.97 \times 10^{-4}$ | $3.29 \times 10^{-4}$ | 1.532 | 4.29 | 43.08 | 40.85 |
| BOTSWANA | $2.34 \times 10^{-4}$ | $2.29 \times 10^{-4}$ | 1.92 | 3.21 | 42.33 | 42.42 |

## 4. Conclusions and Future Works

The human brain is far more complicated and functionalized than current artificial neural networks, and there are still many things we can learn from it. The brain not only can solve a classification or regression problem but, also, can memorize an image, sound, or the feeling of a touch. In this paper, we proposed the storage of a HSI via a neural network. We generated the HSI from a random normally distributed latent code and a neural network. We proved that the random normal distribution provides the most entropy and, thus, is the best distribution for our case. Then, the well-trained network becomes the representation of the HSI. The compression quality and ratio are directly controlled by the structure of the neural network. Since the neural network consists of many near-zero weights when the L1 norm is used as a measure, it occupies a large amount of storage space. With a pruning technique, those weights can be eliminated without much impact on the accuracy, and the compression ratio can be further improved at a large scale.

This paper proposes a novel compression method based on a neural network, and it opens many new directions for further research. We introduced three possible directions that could be further studied. For example, we believe that the compression ratio can be improved by adding some other structures. Furthermore, because the neural network can be treated as a representative of the HSI, many tasks such as classification could be done directly using the GNN, where the layer of the GNN could be manipulated to fit the classification label.

**Author Contributions:** Conceptualization, C.D. and L.Z.; methodology, C.D.; software, C.D.; validation, C.D.; formal analysis, C.D.; investigation, C.D.; data curation, C.D.; writing—original draft preparation, C.D.; writing—review and editing, C.D. and Y.C.; visualization, C.D.; supervision, Y.C.; and project administration, L.Z. All authors have read and agreed to the published version of the manuscript.

**Funding:** This research was funded by the National Key Research and Development Program of China, Project No. 2017YFC1500900, the Natural Sciences Foundation of China, grant number 4197071442, the International Programs & Strategic Innovative Programs, grant number 2017YFE0194900, and the Natural Sciences Foundation of China, grant number 41977154.

**Acknowledgments:** We thank Kimberly Moravec from Liwen Bianji, Edanz Editing China (www.liwenbianji.cn/ac) for editing the English text of a draft of this manuscript.

**Conflicts of Interest:** The authors declare no conflicts of interest.

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
