# Peer review of "Learning-Based Hyperspectral Imagery Compression through Generative Neural Networks"

_remotesensing, doi:10.3390/rs12213657_

Round 1

Reviewer 1 Report

Most of the comments that I pointed out in my previous reviews have now been satisfactorily addressed by the authors.

Also, the presentation quality of this revised manuscript has greatly improved. Nonetheless, there are still some minor aspects that should be addressed. For example, some figures and tables are still miss-positioned and most references are not being properly displayed. Nevertheless, I believe that such minor issues can be easily overcome if the authors perform a very careful final revision of the manuscript. To assist the authors in such procedure, in attach tho this revision I am sending an annotated PDF file.

In conclusion, I find that the manuscript can now be accepted for publication, provided that the above mentioned recommendations are properly implemented.

Author Response

Really appreciate for your comments and I have corrected all the mistakes. For the reference error. I believe something happened when the Word transform to PDF and the cited references becomes an error. Therefore, I submit this PDF version this time and thank you again!

Reviewer 2 Report

All the comments that I have highlighted were satisfactorily addressed by the authors.

However, I have found some minor problems related to the references. In particular, in several points appear of the paper appear the following error:

"[Error! Reference source not found.]"

I would suggest a further proof-reading to correct this issue and some typos, as for instance, the following ones:

- "MSE(After pruning)" -> "MSE (After pruning)" (Table 2)

- "Size of GNN(After pruning)" -> "Size of GNN (After pruning)" (Table 2)

- "CR(After pruning) -> CR (After pruning) (Table 2)

Author Response

(The authors gave the same response as above.)

Reviewer 3 Report

The paper is written, with missing references, low-quality figures, therefore, with low readability. 

Author Response

The missing references have been corrected and the low-quality figures may be caused by the transformation issue from Word to PDF or compressed issue during the upload process. Please check again for this Word version.

Round 2

Reviewer 3 Report

The paper brings some interesting results, which can be published. 

This manuscript is a resubmission of an earlier submission. The following is a list of the peer review reports and author responses from that submission.

Round 1

Reviewer 1 Report

Most of the comments that I pointed out in my previous review were satisfactorily addressed by the authors. Nevertheless, several key issues still remain to be properly presented and/or discussed, namely:

  • The design and architecture of the considered GNN is still poorly presented, including the compression phase and the latent code generation procedure;
  • The experimental evaluation still considers a single dataset, which may put into question the quality of the results provided by the proposed technique;
  • The computational performance and the implementation requirements of the proposed technique are still not discussed.

In what concerns the presentation, I find that the quality of the revised manuscript has greatly improved. However, there are still some aspects that must be addressed. In particular, sections 1, 2.1 (the initial part), 2.1.2 and 2.1.3 are still somewhat confusing. Moreover, some equations, figures and tables are still miss-positioning and now miss-referenced as well. In addition, the list of references should use the same styles to present all the items. Regarding the English presentation, there are still some clerical errors and several grammar issues that must also be addressed.

In conclusion, I find that the manuscript must be further improved before it can be accepted for publication.

Reviewer 2 Report

In this manuscript, the authors introduce a novel approach for the compression of hyperspectral images. More precisely, for the compression process a generative neural network (GNN) is considered.

In my opinion, the authors have worked a lot to improve the paper.

In particular, the authors have significatively improved the "Introduction" section, by emphasizing the novelty of the work and several other aspects related to their proposal.

In general, all the parts are essentially clear. But, I think that the authors should improve the description of Figure 2 and should also improve Section 2.1.2, since it is not entirely clear.

In addition, a further proof-reading is highly suggested, to correct some typos, as for instance, the following ones:

- "training[16]" -> "training [16]" (Page 11)

- "provides a higher entropy than any other distribution.2.1.3. Model weight pruning"  (Page 6)

Reviewer 3 Report

I had reviewed this article before. I see that there is some improvement in this revised version.  Comparative result with the two other methods have been added, quality of the figures has been improved. However, many of my comments and concerned from the previous review have not been addressed; importantly, the structuring of the paper, readability of the main idea such as generation of latent code, how VAE works, calculation of compression ratio, not much discussion of the results, etc.